# First Molecular Characterization of *Cryptosporidium* spp. in Patients Living with HIV in Honduras

**DOI:** 10.3390/pathogens10030336

**Published:** 2021-03-13

**Authors:** Sergio Betancourth, Osman Archaga, Wendy Moncada, Vilma Rodríguez, Gustavo Fontecha

**Affiliations:** 1Microbiology Research Institute, Universidad Nacional Autónoma de Honduras, Tegucigalpa 11101, Honduras; sergiobet94@gmail.com (S.B.); osman.archagav@gmail.com (O.A.); 2Servicio de Atención Integral para Pacientes que Viven con VIH/SIDA-Instituto Nacional Cardiopulmonar, Tegucigalpa 11101, Honduras; karely.mn@gmail.com (W.M.); vil_crl@yahoo.com (V.R.)

**Keywords:** *Cryptosporidium* spp, *C. parvum*, *gp60*, 18S ribosomal, *cowp*, HIV/AIDS, Honduras

## Abstract

Cryptosporidiosis is one of the most important causes of gastroenteritis in the world, especially in low- and middle-income countries. It is caused by the Apicomplexan parasite *Cryptosporidium* spp., and mainly affects children and immunocompromised people, in whom it can pose a serious threat to their health, or even be life threatening. In Honduras, there are no data on parasite species or on molecular diversity or *Cryptosporidium* subtypes. Therefore, a cross-sectional study was conducted between September 2019 and March 2020 for the molecular identification of *Cryptosporidium* spp. in 102 patients living with HIV who attended a national hospital in Tegucigalpa. Stool samples were analyzed by direct microscopy, acid-fast stained smears, and a rapid lateral flow immunochromatographic test. All samples that tested positive were molecularly analyzed to identify the species and subtype of the parasite using three different markers: *gp60, cowp,* and 18Sr. PCR products were also sequenced. Four out of 102 samples (3.92%) were positive for *Cryptosporidium*
*parvum*, and all were assigned to subtype IIa. These findings suggest a possible zoonotic transmission in this population.

## 1. Introduction

New HIV infections worldwide have decreased by 40% since the peak in 1998. However there are currently more than 38 million people living with HIV (PLHIV) with 1.7 million new infections occurring in 2019 [1]. Low- and middle-income countries (LMICs) show only a modest achievement in reducing the burden of this disease due mainly to limited antiretroviral (ARV) coverage and lack of access to health services. In Honduras, more than 23,000 PLHIV were reported in 2019, and although the number of deaths from AIDS has decreased significantly in the country, the number of people newly infected with HIV has remained stable in the last decade [2]. 

People living with HIV/AIDS suffer from numerous and frequent opportunistic infections, many of which pose a serious threat to their health and are life threatening. Cryptosporidiosis is one of the most common causes of gastroenteritis in immunocompromised people [3,4]. The etiologic agent, *Cryptosporidium* spp., is an Apicomplexan protozoan with opportunistic behavior. In healthy, immunocompetent people, it usually causes an asymptomatic infection, or it is the cause of self-limited intestinal symptoms. Conversely, in immunocompromised people, it causes severe diarrhea, fever, nausea, vomiting and dehydration that can lead to the death of the patient. The infection can also progress to chronic diarrhea [5], which can persist for several months in patients with CD4 counts below 50–100 cells/mm^3^ [6,7]. These infections pose an additional challenge for the clinician and the patient because treatment in PLHIV has low efficacy.

The *Cryptosporidium* mechanism of transmission can be either direct or indirect. The former is fecal–oral, through the ingestion of infectious oocysts shed by infected hosts; the latter is through the ingestion of oocysts present in contaminated water or food. Oocysts are highly resistant to natural environmental conditions and even to the chemical treatments used to make drinking water potable, such as chlorine [8]. Since the parasite has the ability to infect a wide range of vertebrate hosts, including humans and farm animals, especially livestock, human infections can have anthroponotic or zoonotic origins [9,10]. In LMICs, cases of cryptosporidiosis are frequent in children under 5 years of age and transmission is believed to follow mostly an anthroponotic pattern. In contrast, outbreaks of zoonotic origin are most frequent in industrialized countries [11]. 

A growing number of *Cryptosporidium* species have been reported to cause infection in humans and other vertebrate hosts [12,13]. The two most frequently reported species in humans and among PLHIV are *C. hominis* and *C. parvum* [14,15], but many other zoonotic species are also commonly found among immunocompromised patients [13,16]. Some authors suggest that the severity of the clinical manifestations of cryptosporidiosis could be associated with different species [17]. Furthermore, *C. hominis* has been associated with a more anthroponotic transmission, while *C. parvum* and other species are primarily associated with zoonotic transmission, although to a lesser extent they can also be transmitted between humans [18]. Hence, it is important to determine the parasite at the species level, either in a geographic region or between specific biological populations. Since identification of the parasite species cannot be done by routine clinical laboratory methods (e.g., modified Ziehl–Neelsen or any acid-fast stain), it is necessary to resort to genetic methods based on molecular markers such as the *gp60* gene or the sequencing of the 18S ribosomal fragment. Not only these techniques allow species identification, but also serve to classify each species into families and subtypes [19,20,21]. 

Studies determining the prevalence of cryptosporidiosis in PLHIV in the Central American region are very scarce, and to our knowledge no previous studies are available genotyping the parasite in Honduras. Therefore, in this study, we investigated for the first time *Cryptosporidium* species and genotypes in HIV-infected individuals in Honduras to increase understanding of *Cryptosporidium* epidemiology in the country. 

## 2. Results

### 2.1. Clinical and Epidemiological Data

One hundred and two adult patients living with the human immunodeficiency virus (PLHIV) with or without diarrhea and seeking routine medical care were recruited for this study. All patients voluntarily agreed to participate after providing their written informed consent. Sixty participants were male (58.82%) and 42 were female (41.18%). The average age was 44.02 years, in an age ranging from 24 to 66 years. Most of the participants (68.6%) resided in the department of Francisco Morazán, where the Honduran capital, Tegucigalpa, is located, while 32 participants came from 12 other departments of the country (Figure 1). With regard to drinking water, the majority (55%) of the participants indicated that they drank bottled water on a regular basis and 40% reported drinking tap water (Figure 2).

The majority (86.3%) of the participants were on antiretroviral treatment at the time of recruitment. Diarrhea in the past week was reported by 29 participants (28.4%). Of these, 16 patients reported having symptoms for less than a week, while 13 patients reported a longer duration. Patients with diarrhea reported an average of five bowel movements per day, with a range between one and 15. Other less common clinical symptoms included fever (25.49%), cough (27.45%), nausea (10.78%), and abdominal pain (16.66%) (Table 1).

Ninety-four (92.15%) participants had a recent count of CD4+ T cells in their medical record. The average number of CD4 cells was 266/mm^3^ (SD = 283; and range 3 to 1000). The mean CD4 lymphocyte count among patients with diarrhea was 239 cells/mm^3^ (range, 7 to 921), while the mean for patients without diarrhea was 310 cells/mm^3^ (range, 3 to 1000). The average time since patients were diagnosed as HIV positive was 6.8 years (SD = 6 years; and range 2 days to 20 years).

### 2.2. Coproparasitological Analysis

All participants submitted a stool sample for coproparasitological analysis. The consistency of most of the samples was soft or well formed (81.4%), while 18.6% of the stools were watery. Only 26.5% of the samples presented mucus in the macroscopic analysis, and none showed the presence of blood or macroscopic parasites.

To evaluate the presence of oocysts, samples were processed with the modified Ziehl–Neelsen stain (Kinyoun) and with an immunochromatographic rapid diagnostic test (RDT) used for the simultaneous detection of *Cryptosporidium* spp. and *Giardia intestinalis*. The Kinyoun stain revealed that four patients out of 102 (3.92%) were infected with *Cryptosporidium* spp. and one patient was infected with *Cystoisospora belli* (0.98%). The RDT also showed the presence of cryptosporidiosis in the same four patients. Two cases of giardiasis in another two patients were also detected by the RDT. The concordance between the Kinyoun stain and the RDT for the diagnosis of cryptosporidiosis was 100%.

Other intestinal infections were detected in 32 participants (Table 1 and Figure 3). Most infections were caused by commensals protozoa, but five potential pathogenic parasites were identified: *Giardia intestinalis*, *Ascaris lumbricoides*, *Trichuris trichiura*, *Cystoisospora belli* and *Entamoeba histolytica/dispar*. Four patients were polyparasitized by two or three microorganisms. Two of the four patients with *Cryptosporidium* were coinfected with *Blastocystis hominis* or *Iodamoeba butschlii*.

### 2.3. Clinical and Laboratory Data of Four Patients Infected by Cryptosporidium spp.

Three of the four *Cryptosporidium*-infected patients were women. All were residents of Tegucigalpa; three indicated that the source of drinking water was bottled water and one drank tap water. Three of them had diarrhea at the time of collecting the fecal sample; three were on antiretroviral therapy and one had a fever. The CD4+ cell range for those four patients was 18–185/mm^3^ (Table 2). 

### 2.4. Molecular Analysis

Three molecular markers were used for the diagnosis of *Cryptosporidium* species: the *gp60* gene, *cowp* gene and the 18S ribosomal fragment. The amplification by nested PCR of the *gp60* gene is capable of differentiating *C. parvum* and *C. hominis* by means of a size polymorphism (Figure 4a) [19]. The four samples showed the presence of *C. parvum* with a product of 848 bp. The isolates belong to the IIaA15G2R1 subtype, with 15 TCA triplets and 2 TCG triplets in the trinucleotide repeat region, one ACATCA repetitive sequence, and a conserved region that matches IIa [21,22,23]. The *cowp* gene was also amplified through nested PCR and subsequently digested with the restriction enzyme RsaI to differentiate both species of *Cryptosporidium* [24]. All four samples revealed a restriction pattern of 413 bp and 129 bp, which was consistent with *C. parvum* (Figure 4b,c). The third molecular marker was the 18S ribosomal gene [10] (Figure 4d). The amplified region of this locus does not show size polymorphisms that could help to differentiate between parasite species and was therefore necessary to sequence the PCR product.

The amplification products of the three genes were sequenced to confirm the parasite’s species. When analyzed with the NCBI BLAST tool, all sequences from the four positive participants confirmed that the species was *C. parvum*. All sequences showed a maximum nucleotide identity of 98.8 % with nucleotide sequences deposited in the GenBank. No nucleotide differences were found between similar sequences. All sequences were deposited in the GenBank database under accession numbers MW493664-7 (18Sr), MW561213-6 (*cowp*) and MW561217-20 (*gp60*).

## 3. Discussion

Intestinal infections by *Cryptosporidium* spp. are the leading cause of chronic diarrhea in immunocompromised people [5,25,26,27] and represent a threat to the lives of the population living with HIV [28,29]. In this study, the presence of *Cryptosporidium* spp. was investigated in a population of HIV-positive adults and a prevalence of 3.92% (4/102) was found. Several studies have aimed to estimate the prevalence of cryptosporidiosis in immunocompromised populations in different countries [5,30,31,32,33,34,35,36]; however, the results are heterogeneous even with data collected within the same country [7,37,38]. The discrepancies between the studies could be attributed in part to their methodological design; however, all the components of the epidemiological triad, which includes the interaction between the host, the parasite and the environment, could be behind these inconsistencies. 

Some authors report a prevalence similar to that found in this study. In a cross-sectional study conducted among 346 immunocompromised patients in Iran, the prevalence of cryptosporidiosis among HIV/AIDS patients was 4.6% and the overall prevalence in the study population was 3.5% [39]. Another study carried out in Indonesia among adult HIV/AIDS patients with chronic diarrhea and without antiretroviral therapy showed a prevalence of 4.9% [38]. Due to the heterogeneity of the information available in the literature on the prevalence of cryptosporidiosis in PLHIV, it is difficult to explain the relatively low prevalence found in this study. However, it could be attributed to the availability of antiretroviral therapies and the consumption of potable drinking water in most cases. Future studies should be carried out to overcome some limitations of this study, such as the small sample size and, therefore, the impossibility of associating cryptosporidiosis with the CD4 count and symptoms of the patients.

The diagnosis of cryptosporidiosis is usually made by microscopic observation of oocysts in the stool after staining with acid-fast methods. More technically complex methods such as ELISA and quantitative PCR have been proposed as an alternative for routine diagnosis [33,40,41], but microscopy remains the gold standard, especially in low- and middle-income country (LMIC) settings. In this study, all samples were analyzed by the Kinyoun stain and an immunochromatographic lateral flow assay (rapid diagnostic test, RDT). Both methods had the same results on the same samples with 100% agreement. A recent cross-sectional study conducted among children with diarrhea from four African countries compared this RDT with a diagnostic panel that included PCR and qPCR tests as gold standards. The authors concluded that the RDT had a very low sensitivity compared to molecular approaches [42]. On the other hand, other authors compared two RDTs against PCR as the reference method in patient samples from three hospitals in Madrid, finding 100% sensitivity and specificity in the RDT used in our study [43]. Although the number of samples analyzed in the present study was small, it is possible to suggest that RDTs could be a good method to be used routinely for the detection of *Cryptosporidium* spp. due to their speed and ease of procedure. However, it is also necessary to consider the moderately higher cost of laboratories in Honduras.

Other intestinal parasites were also reported in this study. More than 31% of the participants revealed at least one parasite, commensal or pathogen. A previous study that included 133 people with HIV in Honduras reported an overall prevalence of intestinal parasites of 67% [44]. *Cryptosporidium* was not found on that occasion. Another study conducted in 1998 among 52 PLHIV and 48 HIV-negative people in San Pedro Sula, Honduras, reported that 55% of all participants and 49% of HIV-positive patients harbored at least one intestinal parasite. These authors found that *Cryptosporidium* spp. and *Strongyloides stercoralis* were present exclusively among people with HIV, while the prevalence of other extracellular parasites was significantly higher in the comparison group [45]. Despite the high prevalence of intestinal parasites reported in these studies, there do not seem to be significant differences with respect to the general population of the country, whose inhabitants, particularly children, have historically reported high prevalences of intestinal parasitism [46,47,48,49,50]. 

To date, more than 20 species have been described, but *C. hominis* and *C. parvum* are the two most common in humans [14,15]. In this study, all four cases were caused by *C. parvum*. In addition to the high heterogeneity of the data found in the literature on the prevalence of cryptosporidiosis in PLHIV, there are discrepancies regarding the most common species that cause infection in this population. Most studies report *C. hominis* as the most prevalent species [13,36,39,51,52,53,54], while in other studies *C. parvum* is the most frequent [16,55,56,57,58]. This discrepancy does not appear to be associated with a geographical distribution, since both groups of reports include studies carried out on three or four continents. Given that *C. hominis* is a more selective parasite and *C. parvum* is more generalist, an anthroponotic transmission versus a zoonotic transmission seems to be the most accepted argument among the authors to explain the phenomenon [9,11,18]. 

Some studies have suggested that *Cryptosporidium* genotypes could influence the epidemiology and severity of symptoms [17], and the hypervariable marker *gp60* has become the most widely used to genotype and classify the parasite [11]. Following this classification, the four isolates in this study belong to the IIaA15G2R1 subtype. This result was corroborated by the three molecular markers, which shows that all can be used successfully to genotype the parasite, although it is preferable to use *gp60* in the future to generate data that better adhere to world trends. An interesting result of this study is that all four cases of cryptosporidiosis were caused by the same subtype of the parasite, although there was no epidemiological link between the patients. Future studies with a larger sample and a “One Health” approach with a multidisciplinary team could shed light on the genetic diversity of *Cryptosporidium* in Honduras [59]. Since this is the first report of the parasite genotypes in Honduras, it is not possible to compare our result with similar reports. However, the genotype IIaA15G2R1 of *C. parvum* has been reported in all continents and is the one with the greatest distribution in both cattle and humans [15,19,21]. It has also been described as the hyper-transmissible subtype [60]. These data would support the hypothesis of a zoonotic transmission among the participants of this study. 

## 4. Materials and Methods

*Study population and sample collection*. This was a cross-sectional study in adult patients living with HIV who attended routine follow-up medical care at the Infectious Diseases department of the Instituto Nacional Cardiopulmonar (INCP), from Tegucigalpa Honduras. This hospital was selected because it specializes in the healthcare of PLHIV/AIDS, both in outpatient services and in hospitalization. The selected patients were over the age of 21 years old and enrolled between September 2019 and March 2020. All patients who agreed to participate voluntarily, regardless of their health status, were included in the study. A questionnaire was applied to collect demographic information that included: name, sex, age, place of residence, source of drinking water, clinical symptoms that included gastroenteritis, daily bowel movement frequency, fever, cough, nausea, vomiting, and abdominal pain. In addition, the last CD4+ T lymphocyte count, antiretroviral treatment status, and duration of HIV infection were extracted from patient’s medical records. Each patient had to provide a single stool sample the same day they were enrolled. The physician provided a sterile container for collection of the stool sample. The samples were transported to the laboratory of the National Autonomous University of Honduras (UNAH) for analysis the same day the patients were enrolled.

*Ethical consideration*. Informed consent was obtained from all patients who voluntarily decided to participate in the study. Participants’ personal information was encrypted to anonymize the stool samples and registration documents. This study was approved by two institutional ethics boards: the Ethics Committee of the INCP (0-21 CE-INCP-2019) and the Ethics Committee of the Maestría de Enfermedades Infecciosas y Zoonóticas-UNAH (03-2019).

*Coproparasitological analysis, Kinyoun stain and rapid diagnostic test.* A single stool sample was obtained from each participant and analyzed the same day as collection. All samples were examined by light microscopy, first with saline and iodine solutions in search of any intestinal parasite, and secondly on a stained smeared slide using a modified Ziehl–Neelsen acid-fast stain (BK Kinyoun Kit, Química Analítica Aplicada, Tarragona, Spain) for the specific detection of coccidian oocysts. Further, a commercial one-step rapid lateral flow immunochromatographic test was performed (CerTest Crypto + Giardia combo card test, BIOTEC S.L., Zaragoza, Spain) according to the manufacturer’s instructions for the simultaneous detection of *Cryptosporidium* spp. and *Giardia* spp. All samples that were positive for *Cryptosporidium* spp. under microscopic analyses or rapid diagnostic testing were stored at −20 °C for further DNA extraction.

*DNA extraction.* Frozen samples were thawed at room temperature for DNA extraction. Genomic DNA was extracted using the QIAamp DNA Stool Mini Kit (Qiagen, Düsseldorf, Germany) following the manufacturer’s instructions. A 200 μL aliquot of extracted DNA was stored at −20 °C for further use.

*Molecular markers.* Positive samples with *Cryptosporidium* oocysts demonstrated by Kinyoun or the RDT were subjected to molecular analysis. The molecular identification of *Cryptosporidium* species was performed using the following three markers: the *gp60* gene, *cowp* gene, and the 18S ribosomal fragment (Figure 5).

*Amplification of the gp60 gene.* A nested PCR was performed for the amplification of a fragment of the *gp60* gene using primers and cycling conditions as described by Roelfsema et al. [19] with some modifications. Both reactions were performed in 25 μL total volume that contained 12.5 μL of GoTaq^®^ Green Master Mix (Promega, Madison, WI, USA), 1.0 μL of each primer at a 10 μM concentration, nuclease-free water, and 1–2 μL of the DNA template. Reactions were amplified by an initial denaturation at 95 °C for 3 min, 35 cycles at 94 °C for 45 s, annealing temperature for 45 s, and 72 °C for 1 min, with a final extension at 72 °C for 10 min. Amplification products from nested reactions were visualized by 1% agarose gel electrophoresis with ethidium bromide. The primer sequences are described in Table 3.

*Amplification and PCR-Restriction Fragment Length Polymorphism (RFLP) of the cowp gene.* For the amplification of the *cowp* gene, a nested PCR was performed with a subsequent digestion of amplicons. Primers and cycling conditions were used as described by Gharieb et al. [24] with some modifications. Both PCR reactions were performed in 50 μL total volume that contained 25 μL of GoTaq^®^ Green Master Mix (Promega, Madison, WI, USA), 2.0 μL of each primer (Table 2) at a concentration of 10 μM, nuclease-free water, and 1–2 μL of DNA template. Both amplification reactions were performed by an initial denaturation at 95 °C for 5 min, 35 cycles at 94 °C for 30 s, annealing temperature for 30 s, and 72 °C for 30 s, with a final extension at 72 °C for 10 min. Amplicons obtained from the second reaction were visualized by 1% agarose gel electrophoresis with ethidium bromide. A 553-bp fragment was expected for both *Cryptosporidium parvum* and *C. hominis.* Next, an RFLP enzymatic digestion was performed using RsaI (Promega, Madison, WI, USA) following the manufacturer’s instructions for 2 h. Fragments obtained from enzymatic digestion were visualized by 1.5% agarose gel electrophoresis with ethidium bromide.

*Amplification of the 18S rRNA locus.* A nested PCR was also performed for the amplification of the 18Sr gene using primers and cycling conditions as described by Al Mawly et al. [10] with some modifications. Reactions were performed in 50 μL total volume that contained 25.0 μL of GoTaq^®^ Green Master Mix (Promega, Madison, WI, USA), 2.0 μL of each primer (Table 2) at a 10 μM concentration, 2.0 μL of Bovine Serum Albumin (BSA) (10 mg/μL), nuclease-free water, and 1–2 μL of the DNA template. Both reactions were amplified by an initial denaturation at 95 °C for 2 min, followed by 35 cycles at 95 °C for 1 min, annealing temperature for 1 min, and 72 °C for 1 min, with a final extension at 72 °C for 10 min. Amplification products from second PCR reactions were visualized by 1% agarose gel electrophoresis with ethidium bromide.

*Sequencing and sequence analysis.* The amplification products of the three markers were sequenced on both strands using the same primers used for the second round of amplification in the nested PCR. Purification and sequencing services were provided by Psomagen (https://lims.psomagen.com, accessed on 16 January 2021). The sequences were edited with the Geneious^®^ 9.1.7 software (Biomatters Ltd., Auckland, New Zealand) and were deposited in the National Center for Biotechnology Information (NCBI; https://www.ncbi.nlm.nih.gov, accessed on 23 January 2021). All sequences were submitted as queries to NCBI via the BLAST tool under default parameters to determine percent identity relative to the most similar sequences available in the GenBank nucleotide collection. The sequences were deposited in the NCBI GenBank. The *gp60* gene sequences were classified according to the subfamilies proposed in the literature [21,22,23]. Subtype assignment is based on the number of trinucleotide repeats and other repetitive sequences. Subtype families are named Ia, etc., for *C. hominis* and IIa, etc., for *C. parvum* [61].

## 5. Conclusions

This is the first report of the genetic characterization of *Cryptospordium* spp. at the species and subtype levels in HIV-positive people in Honduras. *C*. *parvum* IIaA15G2R1 was the only subtype detected and suggests a possible zoonotic transmission in this population. Further research on *Cryptosporidium* should focus on animals and other human populations to expand our knowledge of the subject and to clarify transmission routes.

## Figures and Tables

**Figure 1 pathogens-10-00336-f001:**
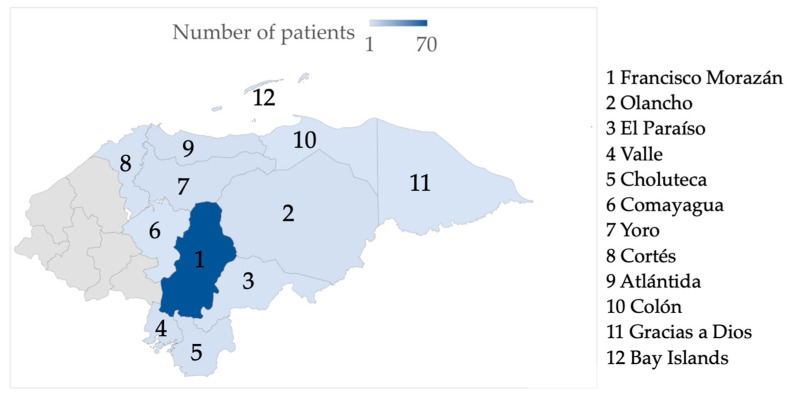
Geographical origin of the study participants.

**Figure 2 pathogens-10-00336-f002:**
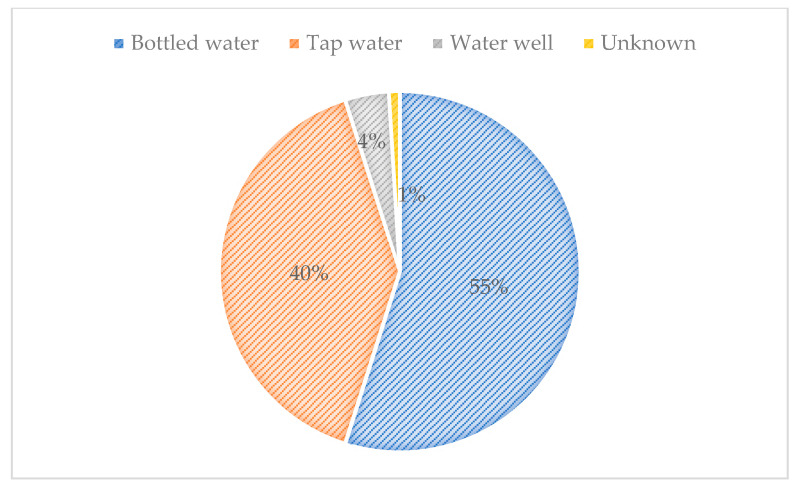
Source of drinking water of the study participants.

**Figure 3 pathogens-10-00336-f003:**
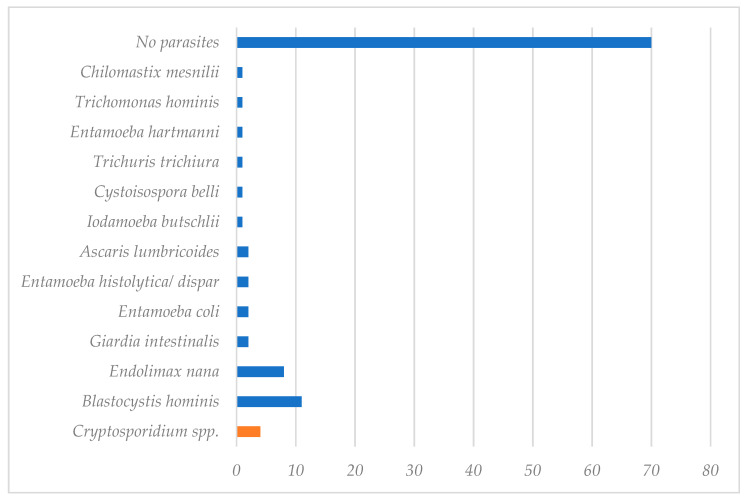
Number and species of intestinal parasites found in the participating individuals.

**Figure 4 pathogens-10-00336-f004:**
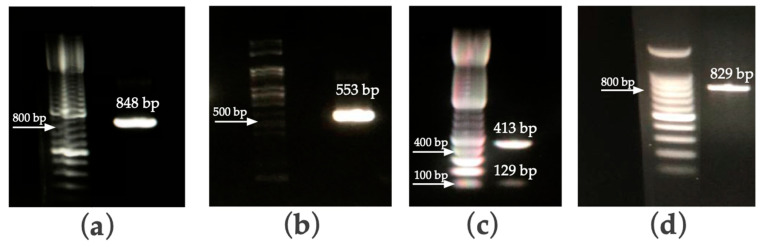
Agarose electrophoresis showing: (**a**) PCR product of the *gp60* gene; (**b**) PCR amplification of the *cowp* gene; (**c**) Digestion of the PCR product of the *cowp* gene using RsaI; (**d**) PCR amplification of the 18S ribosomal fragment.

**Figure 5 pathogens-10-00336-f005:**
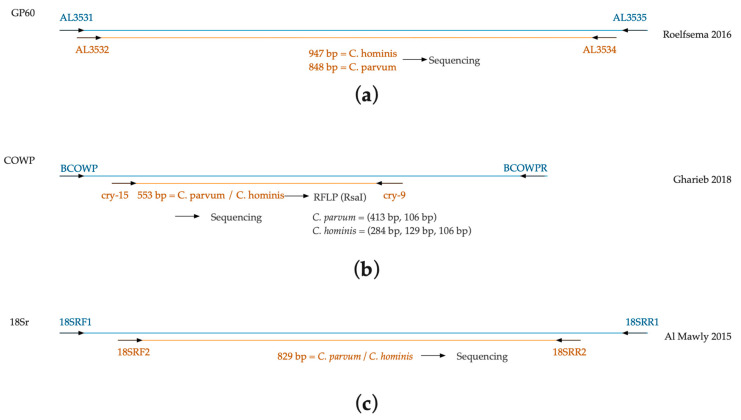
Working algorithm of three molecular markers for the identification of *Cryptosporidium* spp. (**a**) Nested PCR of the gp60 gene [19]; (**b**) Nested PCR of the *cowp* gene, and PCR-RFLP for species identification [24]; (**c**) Nested PCR of the 18S ribosomal gene [10].

**Table 1 pathogens-10-00336-t001:** Clinical and laboratory characteristics of the study population (*n* = 102).

Clinical and Laboratory Findings	Present *n* (%)	Absent *n* (%)	Unknown *n* (%)
Antiretroviral therapy	88 (86.27)	12 (11.76)	2 (1.96)
Diarrhea of at least 3 days	29 (28.43)	73 (71.57)	0 (0)
Fever	26 (25.49)	75 (73.52)	1 (0.98)
Cough	28 (27.45)	73 (71.57)	1 (0.98)
Nausea	11 (10.78)	90 (88.23)	1 (0.98)
Abdominal pain	17 (16.66)	84 (82.35)	1 (0.98)
Mucus in stool	27 (26.47)	75 (73.52)	0 (0)
Macroscopic parasites	0 (0)	102 (100)	0 (0)
White blood cells in stool	2 (1.96)	100 (98.04)	0 (0)
Yeasts in stool	71 (69.61)	22 (21.57)	9 (8.82)
Intestinal parasites ^1^	32 (31.37)	70 (68.62)	0 (0)

^1^ Presence of one or more parasites, including *Cryptosporidium* spp.

**Table 2 pathogens-10-00336-t002:** Demographic, clinical and laboratory data of the four patients with cryptosporidiosis.

Patient	Sex	Age	Drinking Water	Diarrhea	CD4 Cell Count/mm^3^	ARV
1	F	46	Bottled	>3 days	38	Yes
2	F	32	Tap water	No	40	Yes
3	F	35	Bottled	3 days	185	Yes
4	M	44	Bottled	<3 days	18	No

**Table 3 pathogens-10-00336-t003:** Sequences of the primers used for the amplification of three molecular markers.

Gene	Primers for the Nested PCR	Sequence (5′-3′)
*gp60* gene	AL3531F	ATAGTCTCCGCTGTATTC
	AL3535R	GGAAGGAACGATGTATCT
	AL3532F	TCCGCTGTATTCTCAGCC
	AL3534R	GCAGAGGAACCAGCATC
18S ribosomal gene	18SRF1	GTTAAACTGCGAATGGCTCA
	18SRR1	CCATTTCCTTCGAAACAGGA
	18SRF2	CTCGACTTTATGGAAGGGTTG
	18SRR2	CCTCCAATCTCTAGTTGGCATA
*cowp* gene	BCOWPF	ACCGCTTCTCAACAACCATCTTGTCCTC
	BCOWPR	CGCACCTGTTCCCACTCAATGTAAACCC
	Cry-15	GTAGATAATGGAAGAGATTGTG
	Cry-9	GGACTGAAATACAGGCATTATCTTG

## Data Availability

Data are contained within the article.

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
