# Peer review of "First Molecular Characterization of Cryptosporidium spp. in Patients Living with HIV in Honduras"

_pathogens, 2021, doi:10.3390/pathogens10030336_

Round 1

Reviewer 1 Report

This paper report a study of the prevalence of Cryptosporidium in HIV patients from Honduras. Only 4 patients were found positive all with the genotype IIaA15G2 R1 which is the most common C parvum genotype in humans and animals. The number iof cases is too small to conclude about the epidemiology of cryptosporidiosis in Honduras. The only conclusion is the percentage of cryptosporidium infected patients in this cohort of asymptomatic and symptomatic patients. Moreover the RDtest used is of poor quality (Manouana GP et al., 2020). even if the paper is well written , the scarcity of the results obtained do not justify for me a acceptation. 

Author Response

We appreciate the reviewer's comments and agree that the number of detected cases of cryptosporidiosis is low despite the adequate population of participating individuals (n=102). As the reviewer indicates, our results do not allow us to conclude on the epidemiology of parasitosis in Honduras, however the main objective of the work is not epidemiological, nor does it intend to extrapolate the data to the rest of the population. The aim of the study was to determine the species of the parasite for the first time in the country and to classify the parasite according to genotype. This type of study allows starting a line of research that can be expanded later. Regarding the RDT, the reviewer highlights that it is of poor quality. It is true that some authors have shown low sensitivity of the method, however there are also reports that demonstrate the opposite (Gutierrez-Cisneros, et al. Enferm Infecc Microbiol Clin 2011, 29, 201-203, doi:10.1016/j.eimc.2010.09.005.). In any case, the study is not based on RDT but on the technique considered to be the gold standard: microscopy with Kinyoun stain. Furthermore, the concordance in the sensitivity and specificity of the RDT with respect to microscopy was 100% in our study.

Reviewer 2 Report

The study described in this manuscript is the first report of molecular characterization of Cryptosporidium spp in Honduras. Overall, the data is of interest, the manuscript well done, the results clear, the methods suitable for the scope and the conclusions sounding. However, the discussion is too long, with some repetitions and boring. I suggest to shortening it to improve its readability. English needs some attention.

Other concerns

Throughout the manuscript parasite genera and species have to be written in Italic

I suggest using HIV infected or positive than people living on or with HIV

No data is presented about the immunosuppression of the patients included in the study; the data of CD4 counts is no sufficient to state an immunosuppression; I suggest to use HIV infected or immunocopromised individuals

Line 90 and 129: were in antiviral therapy

Line 97: I did not find data of CD4 cells in Table 1

Line 117:  it is not clear if Cryptoporidium infections are included in Table 1

Line 151: parasite species

Fig. 2: what do you mean with well water

Author Response

The study described in this manuscript is the first report of molecular characterization of Cryptosporidium spp in Honduras. Overall, the data is of interest, the manuscript well done, the results clear, the methods suitable for the scope and the conclusions sounding. However, the discussion is too long, with some repetitions and boring. I suggest to shortening it to improve its readability. English needs some attention.

  • We appreciate the reviewer's comments y lamentamos produndamente haberlo aburrido con nuestra larga discusión. Hemos revisado meticulosamente esta sección y aunque hemos eliminado un par de frases, nos es imposible acortarla aun más. La discusión está compuesta por 6 párrafos que coinciden perfectamente con la estructura de los resultados, y cada uno de los párrafos discute un resultado del estudio: (1-2) Prevalencia encontrada, (3) Comparación de técnicas de diagnóstico RDT vs microscopía, (4) Otros parásitos intestinales, (5) especies del parásito and (6) genotipo del parásito.

Other concerns

Throughout the manuscript parasite genera and species have to be written in Italic

  • We have reviewed the entire manuscript and all scientific names are now written in italics. Thank you for this observation and we apologize for the oversight.

I suggest using HIV infected or positive than people living on or with HIV.

  • The formulas "HIV infected or HIV positive" are valid and we welcome the reviewer's suggestion, however, the formula used in this study "people living with HIV" is also widely used in the scientific literature (20,884 articles) and is preferable because it tries to be more delicate with this population and helps to eliminate prejudices against them.

No data is presented about the immunosuppression of the patients included in the study; the data of CD4 counts is no sufficient to state an immunosuppression; I suggest to use HIV infected or immunocopromised individuals.

  • We agree with the reviewer. In this study it has not been possible to confirm a physiological state of immunosuppression among the participants. However, we have never used the term “immunosuppressed” to refer to the participants. The term has always been used to refer to other studies. Either way, the term has been replaced by "immunosuppressed" on all occasions.

Line 90 and 129: were in antiviral therapy

  • It is unclear what the reviewer's suggestion is.

Line 97: I did not find data of CD4 cells in Table 1

  • We greatly appreciate the reviewer's observation. Table 1 does not include that information, and therefore the reference to that table has been relocated.

Line 117:  it is not clear if Cryptoporidium infections are included in Table 1

  • It is right. Thanks for the clarification. The footnote to Table 1 has been modified as follows: Presence of one or more parasites, including Cryptosporidium spp.

Line 151: parasite species

  • Thanks. Done.

Fig. 2: what do you mean with well water

  • A well is an excavation or structure created in the ground by digging, driving, or drilling to access liquid resources. Perhaps the term water well is more appropriate?

Reviewer 3 Report

The manuscript described the results from the prevalence study of cryptosporidiosis among patients living with HIV. Fecal samples were analysed by direct microscopy, acid-fast stained smears and a rapid lateral flow immunochromatographic test. Than positive samples were tested with different markers for identify the species and subtype of parasite.
In my opinion the manuscript is well organized and presented flows/reads very well.

However, few minor things should be improved/considered:

  1. Please consider replacing part from Results (line 77-85) into Material

  1. There was a tendency to use the name of parasites without italic throughout the manuscript, for example:

-line 111: Cryptosporidium spp., Giardia intestinalis

-line 113: Cryptosporidium spp

-line 119 and 120 : Giardia intestinalis, Ascaris lumbricoides, Trichuris trichiura, Cystoisospora belli, Entamoeba histolytica/dispar

-line 121 Cryptosporidium

-line 122 Blastocystis hominis, Iodamoeba butschlii etc.

This is not the correct. Full parasites names should be italic.

Author Response

The manuscript described the results from the prevalence study of cryptosporidiosis among patients living with HIV. Fecal samples were analysed by direct microscopy, acid-fast stained smears and a rapid lateral flow immunochromatographic test. Than positive samples were tested with different markers for identify the species and subtype of parasite.
In my opinion the manuscript is well organized and presented flows/reads very well.

However, few minor things should be improved/considered:

Please consider replacing part from Results (line 77-85) into Material

  • We deeply appreciate the reviewer's suggestion. Indeed, the sex, age, and place of residence of the participants (demographic data) can be interpreted as a result as well as can be included in the methods section. However, this choice is merely subjective and both options are widely found in the literature.

There was a tendency to use the name of parasites without italic throughout the manuscript, for example:

-line 111: Cryptosporidium spp., Giardia intestinalis

-line 113: Cryptosporidium spp

-line 119 and 120 : Giardia intestinalis, Ascaris lumbricoides, Trichuris trichiura, Cystoisospora belli, Entamoeba histolytica/dispar

-line 121 Cryptosporidium

-line 122 Blastocystis hominis, Iodamoeba butschlii etc.

This is not the correct. Full parasites names should be italic.

  • We appreciate the reviewer's kind observation. All scientific names have been written in italics. We apologize for the oversight.